# The Medium-Blocking Discharge Vibration-Uniform Material Plasma Seed Treatment Device Based on EDEM

Yunting Hui, Chen Huang, Yangyang Liao, Decheng Wang, Yong You * and Xu Bai

College of Engineering, China Agricultural University, Beijing 100083, China; hyt@cau.edu.cn (Y.H.);
hc@cau.edu.cn (C.H.); liaoyangyang@cau.edu.cn (Y.L.); wdc@cau.edu.cn (D.W.); baixu@midea.com (X.B.)
* Correspondence: youyong@cau.edu.cn

**Abstract:** Pre-sowing treatment of seeds by plasma can improve seed vigor and promote seed germination and growth. To solve the problems of low processing volume and uneven treatment in plasma seed treatment devices, according to the process scheme of medium-blocking discharge plasma seed treatment, a medium-blocking discharge vibration-uniform material plasma seed treatment device was designed, the structure and working principle of the vibration-uniform material device were systematically analyzed, and the mathematical model of seed force was established. According to electromagnetic vibration theory, the seed sorting and conveying principles were analyzed in the lower trough, and the relevant parameters were selected and calculated. Using EDEM discrete element simulation software, a numerical simulation of alfalfa seed feeding and vibration-uniform material process was carried out. A three-factor, three-level orthogonal test was established. The results showed that the vibration amplitude and groove shape significantly affected the coefficient of variation of seed uniformity on the groove during the seed feeding and vibration-uniform material processes, and the groove wheel speed had a certain effect on the coefficient of variation of uniformity. The main order of factors affecting the uniformity of seed spreading was vibration amplitude B > notch shape C > speed A. The optimal speed was 35 r/min, the optimal notch shape was circular, and the optimal vibration amplitude was 0.55 mm.

**Keywords:** alfalfa seeds; uniform fabric; discrete elements

## 1. Introduction

In recent years, with the improvement of people's living standards and the change of consumption concept, the demand for meat, eggs, and milk has significantly increased [1]. Alfalfa has a high yield and excellent grass quality and is loved by various livestock and poultry [2]. As the main source of protein feed, alfalfa plays an important role in the protein feed shortage in China's animal husbandry [3,4]. Therefore, researching technologies to improve alfalfa yield is of great practical significance.

Seeds are important agricultural production materials, and their vitality affects the entire life cycle of plants. However, the vitality of seeds decreases due to dormancy and physiological aging [5,6]. Effective pre-sowing treatment of seeds is an important way to improve seed viability and to achieve increased crop yields and incomes [7,8]. Currently, commonly used pre-sowing treatment methods for seeds include chemical, biological, and physical treatments [4]. Chemical treatment methods generally use some chemical agents, which can easily damage the environment and even harm human and animal health [9]. Biological treatment methods, such as biological control and plant hormones, are costly and have unclear effects and are also susceptible to environmental influences [10–12]. In recent years, physical treatment methods have become more and more popular due to their environmental protection and good effects [13]. Among them, plasma seed treatment technology is widely used due to its high efficiency, environmental friendliness, and strong applicability [14,15].

Foreign studies have determined that plasma is an effective pre-treatment method for promoting seed germination and growth [16]. Sera et al. [17] observed that, after plasma treatment, the growth of wheat and oat seeds was promoted, and the growth of young roots was accelerated. Moreover, cold plasma has a good sterilizing effect on seeds. Schnabel et al. [18] used medium-blocking discharge and microwave plasma to treat rapeseed seeds, both of which reduced the number of spore rods. Cold plasma can also improve the surface morphology and wetting properties of seeds. Nalwa et al. [19] used low-pressure glow discharge oxygen cold plasma to treat sweet pepper seeds. The surface morphology of the seeds changed and resulted in better seedling characteristics after planting. There have been many studies on plasma seed treatment technology in China. Meng Yiran found that using plasma to treat seeds can promote seed growth, increase yield, enhance seed stress resistance, and kill pathogenic bacteria on grain surfaces [20]. Wang Decheng et al. [21] designed a plasma seed treatment device with temperature control to address the issue of high temperatures caused by continuous treatment with low-pressure radio frequency plasma.

The transportation mode of seeds affects the effect of a plasma seed treatment. The commonly used transportation mode of the sample tray and conveyor belt will lead to an uneven seed treatment effect due to the relatively fixed seed position. Therefore, vibration transportation is widely used because of its simple structure and reliable operation [22]. Xing Jiejie et al. [23] used ADAMS and other software to establish a parameterized electromagnetic vibration orientation device and seed simulation model of corn seeds and verified that the vibration effect of the model was in line with reality from the three aspects of vibration analysis, contact force, and model vibration effect. Xia Hongmei et al. [24] designed a guided vibration seed supply device that can realize uniform seed transportation. Lim [25] pointed toward a possible methodology to model vibrating granular bed systems with inelastic bases using continuum theories. Stepanenko et al. [26] obtained the dependence of the function of the flow rate of grain material on the stepped surface of the vibrating feeder.

Currently, the application of plasma in agriculture, both domestically and internationally, is not comprehensive or well understood in terms of its mechanisms. Most of the developed processing equipment is experimental machines with limited processing capacity. For some batch processing machines that use a conveyor belt as the transportation method, the seeds can block each other, making it difficult to achieve uniform treatment [27–29]. This article aims to address the issues of insufficient processing capacity and uneven treatment of current plasma seed processing machines through motion and dynamic analysis of seeds on a vibrating device. Based on the process plan of medium-blocking discharge plasma seed treatment, we designed a uniform material flow vibration plasma seed treatment device to provide technical support and a theoretical basis for the industrialization of seed treatments.

## 2. Materials and Methods

### 2.1. Establishment of Alfalfa Seed Model

To ensure the accuracy of the simulation results, 100 alfalfa seeds were randomly selected from each, and the length, width, and thickness of the seeds were measured by digital vernier calipers (range 0~150 mm, accuracy 0.01 mm) [30], and the dimensions of the three axes of the seeds were 2.29 mm long, 1.38 mm wide, and 0.95 mm high. Alfalfa seeds are shown in Figure 1. The Poisson's ratio, shear modulus, density, and contact parameters of the alfalfa seeds were obtained through literature research and physical experiments [31,32]. The material properties and contact parameters of the seed discrete element model are shown in Tables 1 and 2 [33,34]. The discrete element model of alfalfa seeds was established using EDEM 2021 simulation software [35], as shown in Figure 2. As the alfalfa seeds are irregular in shape, their discrete element models are difficult to build by individual particles. Therefore, multiple spherical particles are aggregated to build models to simulate the actual seed properties more accurately. The Hertz–Mindlin nonslip model was selected as the particle contact model [36,37].

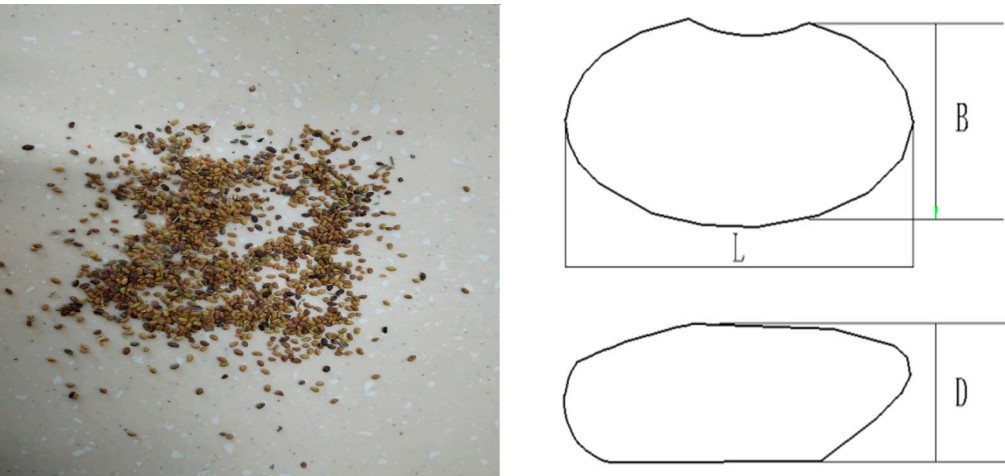

**Figure 1.** Alfalfa seeds. (**Left**): Alfalfa seeds physical picture. (**Right**): Alfalfa seed size diagram ("L" means "long", "B" means wide, and "D" means "high").

**Table 1.** Seed discrete element model and material parameters.

| Parameters | Numerical Value |
|---|---|
| The Poisson's ratio of alfalfa seeds | 0.4 |
| The shear modulus of alfalfa seeds (MPa) | 10 |
| The density of alfalfa seeds $(g/cm^{-3})$ | 0.65 |
| The Poisson's ratio of steel | 0.3 |
| The shear modulus of steel (MPa) | 102 |
| The density of steel $(g/cm^{-3})$ | 7850 |
| The Poisson's ratio of nylon | 0.4 |
| The shear modulus of nylon (MPa) | 90 |
| The density of nylon $(g/cm^{-3})$ | 1200 |

**Table 2.** Contact parameters between seed and material.

| Contact Parameters | Recovery Factor | Coefficient of Static Friction | Coefficient of Rolling Friction |
|---|---|---|---|
| alfalfa-alfalfa | 0.21 | 0.191 | 0.005 |
| alfalfa-nylon | 0.47 | 0.500 | 0.010 |
| alfalfa-steel | 0.63 | 0.075 | 0.023 |

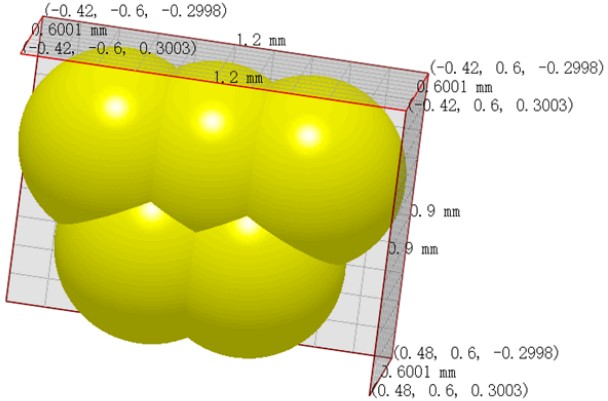

**Figure 2.** Discrete element model of alfalfa seeds.

## 2.2. Device Model

### 2.2.1. Medium-Blocking Discharge Plasma Seed Treatment Device

Technical requirements for the medium-blocking discharge plasma seed treatment device:

(I).   General requirements

The medium-blocking discharge plasma seed treatment device operates under atmospheric pressure conditions, the batch processing volume reached up to 10 kg, the generation power of the medium-blocking discharge ranged from 0 to 500 W (adjustable), and the medium-blocking discharge treatment time for seeds ranged from 5 to 20 s (adjustable).

(II).   Functional requirements

    (a)   The feeding amount was accurate and reliable, and the feeding regulation range was large. Under the condition of striving for precision, the feeding device was structurally designed so that the number of seeds fed each time was accurate, which was conducive to the seeds lying flat and evenly into the treatment bin.

    (b)   Feeding process achieved even spreading of seeds through the device's vibration conveyance. Whether the seeds could evenly enter between the polar plates for media-blocking discharge plasma treatment was one of the key technologies of this device.

    (c)   The transportation was stable, and the operation was reliable. The stable conveying structure prevented the seeds from being affected in the distribution state during the transmission process, thereby improving the stability of equipment performance.

In summary, the main technical parameters of the medium-blocking discharge vibration-uniform material plasma seed treatment device are shown in Table 3.

**Table 3.** The main technical parameters of the medium-blocking discharge vibration-uniform material plasma seed treatment device.

| Name | Numerical Value |
| --- | --- |
| Plasma type | Medium-blocking discharge |
| Plasma generation power (W) | 0~500 |
| Seed processing time (s) | 5~20 (adjustable) |
| Working air pressure (Pa) | $10^5$ (atmospheric pressure) |
| Treatment volume maximum (kg/batch) | 10 (in alfalfa) |
| Temperature (°C) | 15~20 |

According to the design requirements of the medium-blocking discharge plasma seed treatment device, a process plan for a medium-blocking discharge plasma seed treatment device with uniform material flow and vibration was proposed, as shown in Figure 3. The working process of the medium-blocking discharge plasma seed treatment device with uniform material flow and vibration consisted of four parts: the feeding device ensured accurate feeding of seeds; the uniform material distribution device ensured automatic uniform spreading of the seeds during transportation; the conveyor device realized smooth and batch transportation of seeds; and the medium-blocking discharge plasma generation system was used to provide a stable plasma environment for the seeds in the transportation device. The entire process was monitored and controlled in real time by the control system.

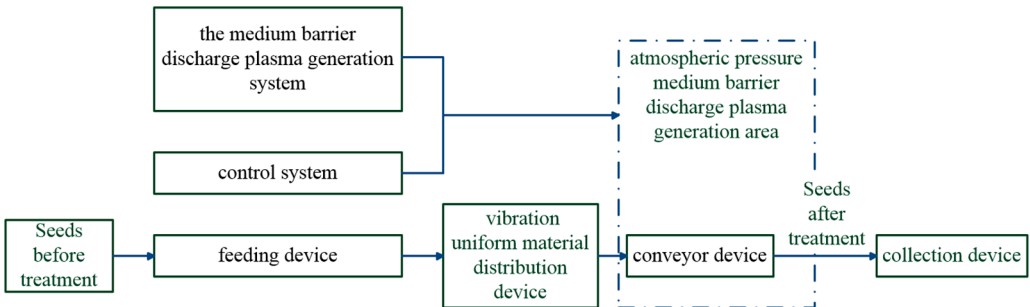

**Figure 3.** Process diagram of medium-blocking discharge plasma seed treatment.

Based on the process plan of the medium-blocking discharge plasma seed treatment device, a medium-blocking discharge plasma seed treatment device with uniform material flow and vibration was designed. The device mainly consisted of a feeding device, a vibration-uniform material distribution device, a conveyor device, a medium-blocking discharge plasma generation system, a collection device, etc. Among them, the feeding device included a feeding bin, seed dispenser body, outer groove wheel, connecting shaft, deceleration motor, etc. The vibration-uniform material distribution device included a groove body, groove body bracket, plate spring, vibrator, clamping iron, connecting fork, base, and other components. The conveyor device included a belt, roller, support frame, and so on. The medium-blocking discharge plasma generation system mainly included an RF power supply, voltage regulator, electrode plate, shielding layer, etc. The collection device mainly consisted of a guiding support frame, collection bin, etc. The structure schematic of the medium-blocking discharge plasma seed treatment device with a vibratory uniform transport function is shown in Figure 4.

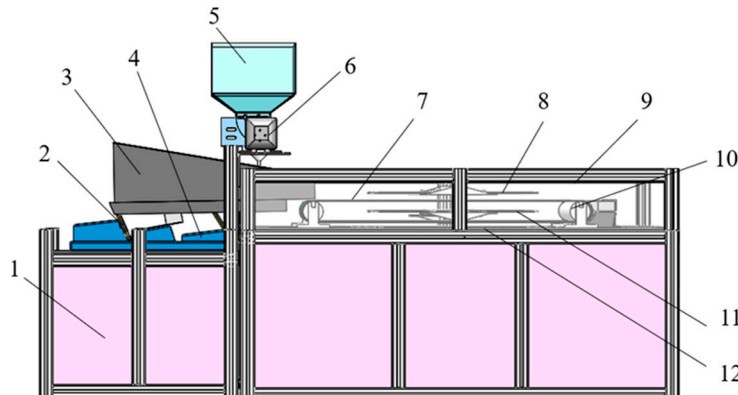

**Figure 4.** The structure schematic of the medium-blocking discharge plasma seed treatment device with a vibratory uniform transport function. 1. Rack; 2. Plate spring; 3. Groove body, Discharger body; 4. Groove body bracket; 5. Material box; 6. Motor; 7. Belt; 8. Upper level board; 9. Upper level board shield; 10. Roller; 11. Lower level board; 12. Lower level board shield.

During operation, the control system controlled the deceleration motor to feed the seeds through the feeding device. The seeds fell onto the vibration-uniform material distribution device, and a unidirectional half-wave rectified current was inputted to the vibrator, causing the vibrator to generate a periodic electromagnetic force that drove the groove body to move back and forth, achieving automatic and uniform spreading of seeds and forward transportation. Finally, the forward-moving seeds fell onto the conveyor belt, where they were transported by the motor-driven belt inside the drum to the processing area for plasma treatment. After treatment, the processed seeds were transported by the conveyor belt to the collection device for collection.

### 2.2.2. Kinematic Analysis of Seed Movement on the Material Groove

The vibration-uniform material distribution device used electromagnetic vibration, and the principle of electromagnetic vibration was to use the periodic electromagnetic force generated by the electromagnetic vibrator as the excitation force to maintain a persistent and stable vibration. The excitation force generated by the electromagnetic vibrator was related to the power supply mode. The power supply mode of the electromagnetic vibrator was mostly unidirectional half-wave rectification, as shown in Figure 5.

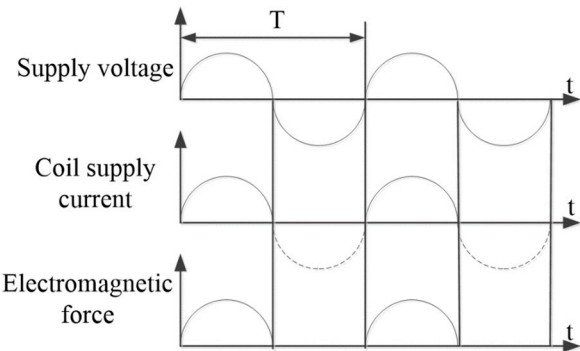

**Figure 5.** Graph of current vs. electromagnetic force.

In order to analyze the movement of seeds on the uniform material distribution device, a force analysis was conducted on the seeds on the material groove. A coordinate system was established with the surface direction of the material groove as the x-axis and the vertical direction of the material groove as the y-axis, as shown in Figure 6.

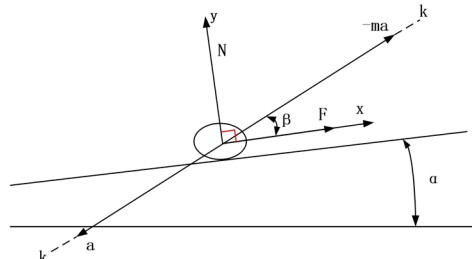

**Figure 6.** Force analysis diagram of seeds on the material groove.

Note: In the diagram, m—the mass of the seed; g—the gravitational acceleration; α—the installation angle of the groove body; β—the vibration direction angle; N—the positive pressure of the groove body on the seeds; F—the frictional force of the groove body on the seeds.

Due to the suction of the electromagnet, the material groove moved from the upper right corner to the lower left corner with the seeds at an acceleration of a, and the seeds were affected by the inertia force of ma. The vertical force to the surface of the material groove, ma sinβ, reduced the positive pressure on the seeds, thereby reducing the frictional force. As shown in Figure 7a, when the parallel force to the surface of the material groove, ma cosβ, was greater than the frictional force F, the seed slid upward along the x-axis; when the vertical force on the seed, ma sinβ, was greater than the gravity force of the seed itself, mg cosα, the seed began to jump and underwent projectile motion. If the throwing time was equal to the descending time of the material groove, the seed had the longest running time and the farthest distance traveled on the material groove when it contacted the groove, as shown in Figure 7b, from point B to C; if the throwing time was less than the descending time of the material groove, the seed would return to the groove earlier and follow the groove down, similar to taking "two steps forward and one step back" on the

material groove, resulting in a smaller displacement of the seed's forward movement, as shown in Figure 7c. If the throwing time of the seed was greater than the descending time of the material groove, the seed would jump higher and return later, but land closer to the groove, as shown in Figure 7d.

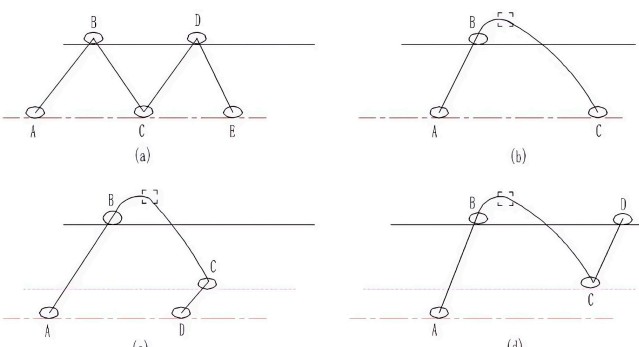

**Figure 7.** The motion states of seeds on the material groove. (**a**)The seed slid upward along the x-axis. (**b**) The throwing time was equal to the descending time of the material groove. (**c**) The throwing time was less than the descending time of the material groove. (**d**) he throwing time of the seed was greater than the descending time of the material groove. "A" indicates the initial position of the seed when the vibration device starts working; "B" indicates the position of the seed when the vibration device reaches the highest point; "C" indicates the location where the seed makes second contact with the material groove.

(1) Displacement, velocity, and acceleration of the material groove.

To further determine the motion of seeds on the groove, a mathematical model was established for the motion of seeds on the material groove. By performing dynamic analysis of the groove performing harmonic motion and a seed on its surface, the displacement formula of the working surface of the groove was derived, thereby determining the displacement, velocity, and acceleration of the groove.

$$S = \lambda sin\omega t \tag{1}$$

$$\omega t = \varphi \tag{2}$$

In the equation: $\lambda$—the single amplitude of the material groove vibrating along the k-k direction; $\omega$—the circular frequency of the vibration; $\varphi$—the phase angle of the vibration; $t$—time.

Decomposing the vibration displacement along the x-axis and y-axis directions, the fractional displacements along the x-axis and y-axis are obtained, respectively, as shown in Equations (3) and (4):

$$S_y = \lambda sin\omega t sin\beta \tag{3}$$

$$S_x = \lambda sin\omega t cos\beta \tag{4}$$

Taking the first derivative and second derivative of time t for Formulas (1), (3) and (4), the velocities $v_x$ and $v_y$ along the x-axis and y-axis, as well as the accelerations $a_x$ and $a_y$ along the x-axis and y-axis were obtained as follows:

$$v_y = \lambda\omega cos\omega t sin\beta \tag{5}$$

$$v_x = \lambda\omega cos\omega t cos\beta \tag{6}$$

$$a_y = -\lambda\omega^2 sin\omega t sin\beta \tag{7}$$

$$a_x = -\lambda\omega^2 sin\omega t cos\beta \tag{8}$$

(2) The conditions for the seed to perform a throwing motion on the material groove.

First, the movement of a single seed on the groove was analyzed. Under this condition, the interaction forces between the seeds were completely neglected. It was assumed that the seed had relative motion with the surface of the material groove, with relative displacements $\Delta\chi$ and $\Delta y$ in the x-axis and y-axis directions, relative velocities $\Delta\dot{\chi}$ and $\Delta\dot{y}$, and relative accelerations $\Delta\ddot{\chi}$ and $\Delta\ddot{y}$.

The force that promoted the sliding of the seed along the x-axis direction was the sum of the inertial and gravitational components of the seed parallel to the surface of the groove, as shown in Formula (9).

$$F_x = -m\left(a_x + \Delta\ddot{\chi}\right) + mg\sin\alpha \tag{9}$$

The positive pressure of the groove surface on the seeds was given by Formula (10):

$$N = -m\left(a_y + \Delta\ddot{y}\right) + mg\cos\alpha \tag{10}$$

In the formula: $m$—the mass of the seed; $\alpha$—the inclination angle of the tank, when conveying downward take the "+" sign, when conveying upward take the "−" sign.

When the seed was in sliding motion relative to the trough surface, it contacted the surface of the trough, with positive pressure $N \geq 0$ and acceleration $\Delta\ddot{y} = 0$. When treated as a throwing motion $N = 0$ and $\Delta\ddot{y} \neq 0$. Formula (11) was obtained.

$$\sin\varphi_d = g\cos\alpha / \omega^2 \lambda \sin\beta \tag{11}$$

In the formula: $\varphi_d$—initial angle of the throwing motion, i.e., the phase angle at the beginning of the throwing motion instantaneously.

$$\varphi_d = \arcsin\frac{1}{D} \tag{12}$$

$$D = K\sin\beta / \cos\alpha \tag{13}$$

$$K = \omega^2 \lambda / g \tag{14}$$

In the formula: $D$—throwing index, indicating the characteristics of the throwing motion; $K$—vibration intensity.

From the above formula, it can be seen that when $D > 1$, there was a solution to the initial throwing phase angle $\varphi_d$, and the seeds could have throwing motion on the surface of the groove; when $D < 1$, $\varphi_d$ had no solution and the seeds could not perform throwing motion on the surface of the groove. Therefore, it could be concluded that the throwing index should not be less than 1 in order to make the seeds do throwing motion on the surface of the groove.

Since the vibration intensity $K = \omega^2 \lambda / g$ and $\omega = \frac{2\pi n}{60}$, after choosing the amplitude $\lambda$, the vibration frequency was calculated according to Formula (15):

$$n = 30\sqrt{\frac{Dg\cos\alpha}{\pi^2 \lambda \sin\beta}} \tag{15}$$

If the vibration frequency $n$ was pre-selected, the amplitude was calculated according to Formula (16):

$$\lambda = \frac{900Dg\cos\alpha}{\pi^2 n^2 \sin\beta} \tag{16}$$

(3) Selection of throwing index D.

The movement state of the seed on the surface of the groove depended on the throwing index D. As can be seen from Formulas (12) and (13), the throwing index D should not be less than 1 in order to make the seeds do throwing movement on the surface of the groove.

The value of D normally ranged from 1 to 3, so that the time of the throwing process was less than the vibration cycle, thus improving the efficiency of the machine.

(4) The inclination angle α of the groove body and the vibration direction angle β.

When the seeds did throwing motion on the groove, from the perspective of improving the conveying speed, there was an optimal vibration direction angle at different trough inclination corresponding to each vibration intensity K. In order to improve the conveying ability of the vibration-uniform material distribution device for the seeds, the groove body was installed with a downward tilt because the inclination angle α was generally between 10° and 15° and was not too large. Therefore, this design selected an inclination angle of 10°. Typically, the value range of K was between 2 and 5, but due to the small size of the seed particles, the K value was relatively large, and the value of K was set to 8 in this design. Thus, the vibration direction angle β can be determined to be 21°.

(5) Vibration amplitude and vibration frequency

Generally, electromagnetic vibrating machines used high frequency and small amplitudes, with amplitudes usually around 3000 times/min and vibration amplitudes of 0.5–1 mm. Due to the small volume of material acted upon in this design, the vibration amplitude was not necessarily too large. Therefore, the vibration frequency was 60 Hz, and the vibration amplitude was 0.5 mm.

(6) Theoretical conveying speed.

The theoretical conveying speed of seeds on the material groove was calculated using Formula (17):

$$v_d = f(D)\omega a cos\beta \tag{17}$$

Since the throwing index D was known, $f(D)$ was determined as 0.92 from the dimensionless coefficient curve, and $v_d$ was calculated to be 0.19 m/s.

(7) Actual conveying speed.

The actual conveying speed of seeds on the material groove was calculated using Formula (18):

$$v_m = \gamma_a C_h C_m C_w v_d \tag{18}$$

In the formula: $\gamma_a$—inclination correction coefficient, $\gamma_a = 0.9$; $C_h$—is the material layer thickness influence factor, which was taken as 0.9 here; $C_m$— the material property influence factor, and since alfalfa seeds belong to granular materials, $C_m$ was selected as 0.9; $C_w$—the sliding motion influence factor, since D = 2.5, the effect of sliding motion was ignored, so it was taken as 1 here.

Finally, the actual conveying speed of seeds on the material chute, $v_m$, was calculated as 0.14 m/s.

### 2.3. Experimental Design

Whether the seeds were spread evenly during the vibrating conveying process on the material groove was an evaluation index for the performance of the feeding device and the uniform vibration conveying device. A "Grid Bin Group" was set in the middle of the material groove to display the number of seed particles passing through the set area, as shown in Figure 8.

Three types of groove wheel shapes were selected, namely circular arc groove, conical arc groove, and right-angled groove. The rotational speeds of the groove wheels were set as 35 r/min, 45 r/min, and 55 r/min. The vibration amplitudes of the material groove were set as 0.45 mm, 0.5 mm, and 0.55 mm. The simulation was carried out under the above conditions, and a three-factor, three-level simulation experiment was conducted using the uniformity variation coefficient as the evaluation index. The factor level table is shown in Table 4.

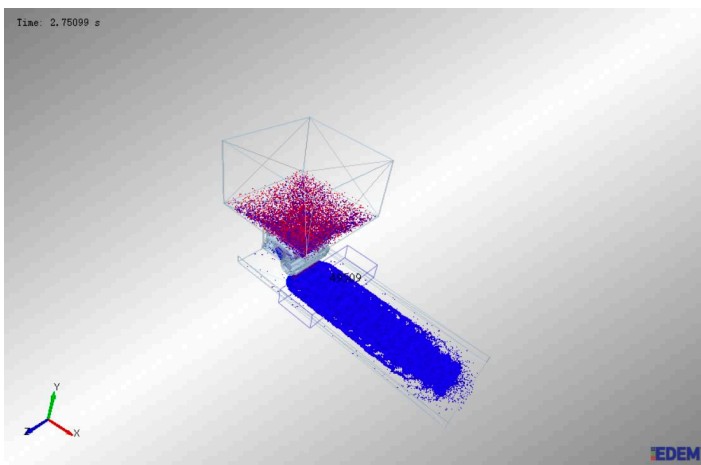

**Figure 8.** Setting of particle flow rate.

**Table 4.** Table of factor levels.

| Level | Factor | | |
|---|---|---|---|
| | **A (r/min)** **Rotational Speed** | **B (mm)** **Vibration Amplitudes** | **C** **Groove Wheel Recess Shape** |
| 1 | 35 | 0.45 | circular arc groove |
| 2 | 45 | 0.5 | conical arc groove |
| 3 | 55 | 0.55 | right-angled groove |

The coefficient of variation was given by Formula (19):

$$Y_z = \sqrt{\frac{1}{n-1}\sum(X_i - x)^2} \times 100\% \qquad (19)$$

In the formula: *X*—the average number of seeds distributed per measurement segment; *n*—the total number of samples in the experiment; *X_i*—the number of seeds per 100 mm measuring section.

## 3. Results and Discussion

The orthogonal table $L_9(3^4)$ was selected to arrange the orthogonal test scheme, and the fourth column was blanked as the error term in the variance treatment. The uniformity variation coefficient was used as the evaluation index to analyze and determine the optimal groove shape in the feeding device and the optimal amplitude parameter in the uniform vibration device. The experimental results are shown in Table 5.

**Table 5.** Orthogonal test results.

| Serial Number | A (r/min) Rotational Speed | B (mm) Vibration Amplitudes | C Groove Wheel Recess Shape | The Coefficient of Variation (%) |
|---|---|---|---|---|
| 1 | 1 | 1 | 1 | 36.18 |
| 2 | 1 | 2 | 2 | 42.72 |
| 3 | 1 | 3 | 3 | 36.06 |
| 4 | 2 | 1 | 2 | 42.72 |
| 5 | 2 | 2 | 3 | 33.91 |
| 6 | 2 | 3 | 1 | 35.49 |
| 7 | 3 | 1 | 3 | 33.84 |
| 8 | 3 | 2 | 1 | 35.51 |
| 9 | 3 | 3 | 2 | 31.48 |

### 3.1. Analysis of Extreme Differences

When using the uniformity variation coefficient as the evaluation index, the extreme difference analysis of the seed feeding vibration conveying simulation orthogonal test is shown in Table 6.

**Table 6.** Table of extreme difference analysis of the feeding vibration conveying simulation orthogonal test.

| Serial Number | A (r/min) Rotational Speed | B (mm) Vibration Amplitudes | C Groove Wheel Recess Shape | The Coefficient of Variation (%) |
|---|---|---|---|---|
| 1 | 1 | 1 | 1 | 37.30 |
| 2 | 1 | 2 | 2 | 27.10 |
| 3 | 1 | 3 | 3 | 28.50 |
| 4 | 2 | 1 | 2 | 45.30 |
| 5 | 2 | 2 | 3 | 37.80 |
| 6 | 2 | 3 | 1 | 21.20 |
| 7 | 3 | 1 | 3 | 60.40 |
| 8 | 3 | 2 | 1 | 28.50 |
| 9 | 3 | 3 | 2 | 24.50 |
| $\overline{K}_{11}$ | 30.97 | 47.667 | 29.00 | |
| $\overline{K}_{12}$ | 34.77 | 31.13 | 32.30 | |
| $\overline{K}_{13}$ | 37.80 | 24.73 | 42.23 | |
| $R_j$ | 6.83 | 3.24 | 13.23 | |
| excellent level | $A_1$ | $B_3$ | $C_1$ | |
| Primary and secondary factors | | B > C > A | | |

When the uniformity variation coefficient was used as the evaluation index, the smaller its value, the better. From Table 6, it can be seen that the primary and secondary factors affecting the uniformity of seed spreading in the feeding and vibrating conveying process were in the following order: vibration amplitude B > groove shape C > rotational speed A. The optimal levels were: the optimal rotational speed was $A_1$ = 35 r/min; the optimal groove shape was $C_1$ = circular arc shape; and the optimal vibration amplitude was $B_3$ = 0.55 mm.

### 3.2. Analysis of Variance

The variance analysis of the seed feeding and vibrating conveying simulation orthogonal experiment is shown in Table 7.

**Table 7.** Analysis of variance for the seed feeding and vibrating conveying simulation orthogonal experiment.

| Source | Sum of Squares Formula | Degree of Freedom | Mean Square | F | Significance-P |
|---|---|---|---|---|---|
| Modified model | 1195.267 [a] | 6 | 199.211 | 27.360 | 0.036 |
| intercept distance | 10,719.15 | 1 | 10,719.15 | 1472.186 | 0.001 |
| Rotational Speed | 70.336 | 2 | 35.168 | 4.830 | 0.172 |
| Vibration amplitude | 840.249 | 2 | 420.124 | 57.701 | 0.017 |
| groove shape | 284.682 | 2 | 142.341 | 19.549 | 0.049 |
| Error | 14.562 | 2 | 7.281 | | |
| Total | 11,928.9 | 9 | | | |
| Total after correction | 1209.83 | 8 | | | |

Note: $p < 0.05$ indicates a significant effect; $p \geq 0.05$ indicates no effect, "a" indicates that the modified model is significant.

From the analysis of variance table, it can be seen that when using the uniformity variation coefficient as the evaluation index, $R^2 = 0.952$, indicating a good fitting effect. Under a confidence level of 95%, the vibration amplitude and groove shape had a significant impact

on the uniformity variation coefficient of seeds on the groove body, while the rotational speed of the groove wheel had no influence on the uniformity variation coefficient.

### 3.3. Uniformity Test

The uniformity test was conducted in the Forage Machinery Laboratory of China Agricultural University. Seeds were fed from the feeding device. The rotational speed of the groove wheel of the feeding device was set at 35 r/min, and the vibration frequency of the vibration-uniform material distribution device was set at 60 Hz. The uniformity test of the plasma seed treatment device was carried out (Figure 9). The samples for the uniformity test were the seeds transported to the discharge area, and random samples were taken within the discharge area with a sampling area of 20 mm × 20 mm. Sampling was repeated 10 times, and each sample was tested twice. The weight of the obtained alfalfa seed samples was measured using an electronic balance (OHAUS Instruments Ltd. Shanghai, China, precision 0.0001 g).

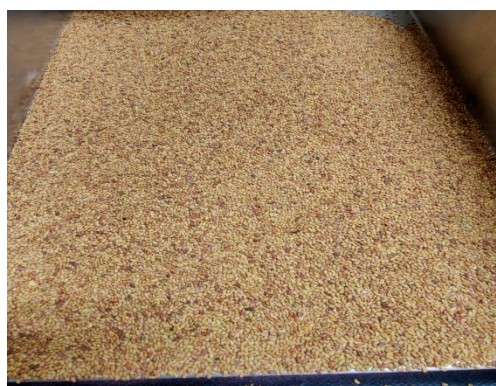 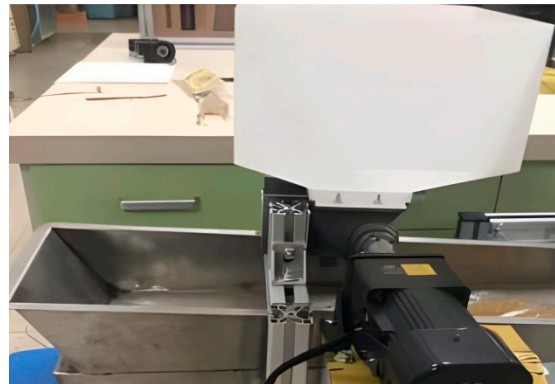

**Figure 9.** The uniformity test of the plasma seed treatment device. **Left**: The uniformity test process. **Right**: The vibration-uniform material distribution device.

According to the uniformity evaluation method of CNAS-GL003:2018 "Guidelines for the Evaluation of Uniformity and Stability of Proficiency Testing Samples", one-way ANOVA was selected to evaluate the uniformity of the seeds when entering the discharge region [38]. The calculation formula is shown below:

$$M = \sum_{i=1}^{m} n_i \tag{20}$$

In the formula: $n_i$—the number of times the $i$th seed sample was tested; $M$—total number of tests for all seed samples; $m$—total number of seed samples to be tested.

The average value of the test for a single seed sample:

$$\overline{x_i} = \sum_{j=1}^{n} \frac{x_{ij}}{n_i} \tag{21}$$

In the formula: $x_{ij}$—the value of the $i$th seed sample tested at the $j$th time; $\overline{x_i}$—test averages for each seed sample.

Total average of all samples $\overline{\overline{x_i}}$ :

$$\overline{\overline{x_i}} = \sum_{i=1}^{m} \frac{\overline{x_i}}{m} \tag{22}$$

Mean square value between samples:

$$MS_1 = \frac{1}{m-1} \sum_{i=1}^{m} (\overline{x_i} - \overline{x})^2 n_i \tag{23}$$

Mean square error within sample:

$$MS_2 = \frac{1}{M-m}\sum_{i=1}^{m}\sum_{j=1}^{n}\left(\overline{x_{ij}} - \overline{x_i}\right)^2 \tag{24}$$

Statistics used to characterize inter-sample variability in one-way ANOVA methods *F*:

$$F = \frac{MS_1}{MS_2} \tag{25}$$

The weighed alfalfa samples were subjected to one-way ANOVA, and the results are shown in Table 8. The F statistic was 0.652. By referring to GB/T 4086.4-1983, the critical value for $F_{0.05(9,10)}$ was found to be 3.02. Compared with the value of F of the statistic, it met the requirement that F was less than the critical value, which indicates that at the 0.05 level of significance, this uniformity of the tested alfalfa seed samples in the discharge region met the requirement of uniformity for proficiency testing.

**Table 8.** Results of one-way ANOVA for alfalfa samples.

| Source of Variance | Degree of Freedom | Sum of Squares | Mean Square | F |
|---|---|---|---|---|
| Between samples | 9 | 0.02821 | 0.003134 | 0.652 |
| Within-sample | 10 | 0.04805 | 0.004805 | |

## 4. Conclusions

(1) The key components of the medium-blocking discharge vibration-uniform conveying plasma seed treatment device were designed, and kinematic analysis of the seed on the groove was conducted to determine the relevant parameters. The operating frequency of the vibration-uniform feeding device was 60 Hz, the vibration amplitude was 0.5 mm, the vibration direction angle β was 21°, the trough inclination angle α was 10°, and the actual conveying speed of the seed on the trough $v_m$ was 0.14 m/s.

(2) Using the discrete element method and based on EDEM software, the effects of the rotational speed of the groove wheel, vibration amplitude, and groove shape on the coefficient of variation of seed uniformity during feeding and vibrating conveying were analyzed. The results showed that the vibration amplitude and groove shape had a significant impact on the uniformity variation coefficient of seeds on the groove body, while the rotational speed of the wheel had no influence on the uniformity variation coefficient. The optimal rotational speed of the wheel was 35 r/min, the best groove shape was the circular arc groove, and the optimal vibration amplitude was 0.55 mm.

(3) Simulation experiments were conducted on the seed feeding and vibration conveyance processes, and the optimal rotational speed of the wheel, groove shape, and vibration amplitude were determined. The uniformity test of the plasma seed treatment device was conducted on a vibration-uniform material distribution device. It was demonstrated that at a significance level of 0.05, the uniformity of the tested alfalfa seed samples in the discharge area met the requirements of proficiency testing uniformity. In the future, prototype manufacturing and processing will be carried out based on these parameters, and the machined prototype will be used for testing to compare and verify the reliability of the simulation results.

**Author Contributions:** Conceptualization, Y.H.; Methodology, C.H.; Formal analysis, Y.L.; Resources, D.W.; Writing—review & editing, Y.Y.; Investigation, X.B. All authors have read and agreed to the published version of the manuscript.

**Funding:** This research was funded by the China Agriculture Research System (CARS-34) and the National Key R&D Program of China (2022YFD2001902).

**Data Availability Statement:** Not applicable.

**Conflicts of Interest:** The authors declare no conflict of interest.

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
