# Peer review of "The Medium-Blocking Discharge Vibration-Uniform Material Plasma Seed Treatment Device Based on EDEM"

_agronomy, doi:10.3390/agronomy13082055_

Round 1

Reviewer 1 Report

Manuscript “The medium-blocking discharge vibration-uniform material plasma seed treatment device based on EDEM”

This manuscript will be interesting for scientists, who are interested in studying the movement of bulk materials on vibrating surfaces. Also, these scientific results will have practical application in agriculture. In order to improve the manuscript, I suggest the following corrections:

1. The results of measuring the length, width and thickness of the seeds should be presented in the article section 2.1.

2. There are no references in article section 2.1 (lines 77, 78, 80).

3. The method of determining contact parameters between seed and material (recovery factor, coefficient of static friction, coefficient of rolling friction), which are given in Table 2, should be specified.

4. The seeds after pre-sowing treatment should be shown on the Process diagram (Fig. 2).

5. Figures 1-7 and Tables 1-6 are not mentioned in the text.

6. The y-axis and the normal reaction N (Fig. 5) must be perpendicular to the surface along which the seed moves.

7. The article states: “The vertical component of the force, masinβ” (page 6, line 155). But this is not the vertical component of the force ma, it is the component of the force ma along the y-axis. The y-axis is not vertical.

8. It needs to be clarified why in Equations (9) and (10) the components  and  have the sign “-”.

9. In the Introduction, an analysis of pre-sowing treatment of seeds methods is carried out. But the material of the article is devoted to the study of the movement of seeds on a vibrating surface and the justification of the optimal kinematic parameters of this surface. Therefore, in my opinion, it is necessary to analyze the results of modeling the movement of bulk materials on vibrating surfaces in the Introduction. Accordingly, the aim of the article needs to be adjusted.

Author Response

Dear reviewer,

Thanks very much for taking the time to review this manuscript. The revised article and the response to reviewer comments can be found in the following PDF file.

Reviewer 2 Report

1. The text lacks an introduction to the details of the device design. The theoretical analysis such as the mathematical model of the seed force analysis is relatively simple, and the simulation results lack the verification of real experiments.

2. The device does not seem to be complicated, is it possible to carry out a real test? If so, I think it is more appropriate for this study to use real experimental research.

3. The source of the selection of material property parameters in Table 1 and Table 2 is unclear and needs further elaboration. Parameters such as the shear modulus of steel seem to be wrong. The accuracy of the simulation results needs to be verified.

4. It is generally believed that: p value<0.01 means extremely significant effect, p value<0.05 means significant effect, p value>0.05 means no significant effect. The choice of P value in this paper is different from the conventional, please explain.

5. I have never seen the marking method of the citation, please check whether it meets the format requirements of the journal.

6. Many citations, charts, etc. are missing in the text; formulas, symbols, etc. are not written in a standardized manner.

Generally poorly written, grammar check should be done for the whole text to improve the clarity and reading. Some long sentences made it difficult to follow the text clearly.

Author Response

(The authors gave the same response as above.)

Reviewer 3 Report

I have gone through the manuscript. Everyone knows that plasma processing technology is mainly used in industries, military, electronics, and other fields. In recent years, it has also been widely used in agriculture.

Plasma seed processing technology involves treating seeds with plasma before sowing. It can significantly improve seed vitality and increase crop yield. The author's research on this technology has great practicality. Below you can find my main comments:

1. The author of this article mentioned in the introduction that the current plasma seed processing equipment has weak processing capabilities. What is the reason for this phenomenon? Is the technical difficulty high or the cost high? What is the existing equipment processing capacity? What is the processing capacity of the author's research equipment? Is there any significant improvement or enhancement?

2. For the convenience of readers, the reference numbers in the manuscript can be in Arabic numerals.

3. I suggest the author add information on the model and manufacturer of the instruments used.

4. This article only conducted simulation experiments on the developed plasma treatment equipment. I am not sure if the author has conducted corresponding physical experiments? Generally speaking, conducting on-site tests can more effectively demonstrate the practicality and reliability of the designed equipment.

Thank you.

Author Response

(The authors gave the same response as above.)

Reviewer 4 Report

Hui Yunting, et al. studied the medium-blocking discharge vibration-uniform material plasma seed treatment device based on EDEM.

 I have some suggestions for revision.

 1.       Please improve the resolutions of all figures.

2.       For Fig. 1, can the authors provide photos of seeds as an example to understand the seed models?

3.      I recommend the authors provide a real photo of the equipment in Fig. 3.

English is okay. 

Author Response

(The authors gave the same response as above.)

Round 2

Reviewer 1 Report

Manuscript “The medium-blocking discharge vibration-uniform material plasma seed treatment device based on EDEM”

I am satisfied with the responses to my questions/issues raised in my initial review. I recommend that the paper be accepted in present form.

Author Response

Dear reviewer,

         Many thanks for your positive comments. It is my great honor to receive your recommendation.

Reviewer 3 Report

The conclusion of this article lacks necessary physical experimental verification and is currently not suitable for publication. It is recommended that the author conduct relevant validation experiments and resubmit the manuscript.

Author Response

Thanks very much for taking the time to review this manuscript. The revised article and the response to reviewer comments can be found in the following PDF file.

Round 3

Reviewer 3 Report

What is Figure 9? Is it a device or an experimental process?

Author Response

Dear Reviewer,

       Thanks very much for taking the time to review this manuscript. The revised article and the response to reviewer comments can be found in the following PDF file. 
